# Examining Innovative Technologies: Nano-Chelated Fertilizers for Management of Wheat Aphid (*Schizaphis graminum* Rondani)

**DOI:** 10.3390/insects15030209

**Published:** 2024-03-20

**Authors:** Masoud Chamani, Bahram Naseri, Hooshang Rafiee-Dastjerdi, Javid Emaratpardaz, Reza Farshbaf Pourabad, Ali Chenari Bouket, Tomasz Oszako, Lassaad Belbahri

**Affiliations:** 1Department of Plant Protection, Faculty of Agriculture and Natural Resources, University of Mohaghegh Ardabili, Ardabil 5619911367, Iran; hooshangrafiee@gmail.com; 2Department of Agronomy and Plant Breeding, Faculty of Agriculture, University of Tabriz, Tabriz 5137779619, Iran; javid_emarat@yahoo.com; 3Department of Plant Protection, Faculty of Agriculture, Ege University, 35100 Izmir, Türkiye; reza.farshbaf.pourabad@ege.edu.tr; 4East Azarbaijan Agricultural and Natural Resources Research and Education Centre, Plant Protection Research Department, Agricultural Research, Education and Extension Organization (AREEO), Tabriz 5355179854, Iran; a.chenari@areeo.ac.ir; 5Institute of Forest Sciences, Faculty of Civil Engineering and Environmental Sciences, Bialystok University of Technology, Wiejska 45E, 15-351 Bialystok, Poland; t.oszako@ibles.waw.pl; 6University Institute of Teacher Education (IUFE), University of Geneva, 24 Rue du Général-Dufour, 1211 Geneva, Switzerland; lassaad.belbahri@unige.ch

**Keywords:** *Schizaphis graminum*, life table, IPM, population growth

## Abstract

**Simple Summary:**

The study investigated how different types of fertilizers affected wheat aphids, a harmful pest for wheat plants. The nano-Cu fertilizer had negative effects on the aphids, leading to shorter lifespans and fewer babies. In contrast, the nitrogen fertilizer was not effective in controlling the aphids. The nano-Fe and nano-Zn fertilizers had fewer negative impacts compared to nano-Cu. While nitrogen treatment had positive effects on wheat aphid lifespan. Overall, the study suggests that incorporating nano-Cu into pest control strategies for wheat aphids could be a promising approach. In conclusion, choosing the right fertilizer can make a big difference in managing pests like aphids on wheat plants, and using nano-Cu could be a valuable tool in protecting crops.

**Abstract:**

The use of nanofertilizers has both advantages and concerns. One benefit is that nano-fertilizers can enhance plant resistance against insect pests, making them a valuable strategy in integrated pest management (IPM). This study focused on the effect of wheat leaves treated with nano-chelated fertilizers and nitrogen (N) fertilizer on the wheat aphid (*Schizaphis graminum* Rondani), a harmful pest of wheat plants that transmits dangerous viruses. The nano-Cu treatment showed the longest pre-adult longevity. Additionally, the nano-Cu treatment resulted in the lowest adult longevity, fecundity, nymphoposition day number, intrinsic rate of population growth (r), finite rate of population increase (λ), and net reproductive rate (R_0_) and gross reproductive rate (GRR). Also, nano-Cu treatment led to the highest amount of (T). The N treatment led to the highest levels of fecundity, nymphoposition days, r, λ, and R_0_. Nano-Fe and nano-Zn demonstrated fewer negative effects on *S. graminum* life table parameters than nano-Cu. Our results indicate that N treatment yielded numerous advantageous effects on the wheat aphid while simultaneously impeding the efficacy of the aphid control program. Conversely, nano-Cu treatment exhibited a detrimental influence on various parameters of the aphid’s life table, resulting in a reduction in the pest’s fitness. Consequently, the integration of nano-Cu should be seriously considered as a viable option in the IPM of the wheat aphid.

## 1. Introduction

Wheat (*Triticum aestivum* L.) is one of the most important food sources for humans, providing essential nutrition and serving as the main ingredient in most processed foods. Additionally, wheat requires less water for cultivation compared to other similar crops [1]. The wheat common bug, or wheat aphid (*Schizaphis graminum* Rondani), is known as one of the most important harmful insects for winter wheat and sorghum [2,3]. The introduction of *S. graminum* as a wheat pest dates back to 1882 in the United States [4]. Although the *S. graminum* outbreak on sorghum was documented in 1916, sorghum was known to be a host for the pest as early as 1863 [4,5]. This insect pest feeds on leaves and stems [6]. The reproductive potential of this aphid is clearly defined and substantial; nevertheless, when subjected to biotic or abiotic pressures, this potential may display notable variations. [2]. Greenbugs can infest some grains and turfgrasses too [7]. In addition to the detrimental effects on wheat plants resulting from sap feeding and weakening, the wheat aphid also serves as a vector for various viral diseases, including barley yellow dwarf virus [8], sugarcane mosaic virus [9], and maize dwarf mosaic virus [10].

Considering that research on *S. graminum* has been ongoing for several years and most reports on its pathogenicity date back to past years, recent studies continue to focus on its control and life cycle, highlighting the significance of this pest. Recent studies have primarily focused on controlling the wheat aphid through various approaches, including the use of *Bacillus subtilis* [11], predators such as ladybirds [12], soil amendments [13], insecticides [14], as well as genetic resistance techniques like QTL mapping [15]. Recently, the use of nano-materials as another component of integrated pest management, aiming to enhance plant resistance and weaken pest infestation, has received significant attention [16].

The utilization of resistant varieties is a conventional and widely employed method for effectively managing plant pests and diseases. However, it is imperative to recognize that not all plants possess inherent resistance to these detrimental factors. Consequently, it becomes necessary to enhance plant resilience through the application of appropriate nutrients, thereby fortifying their defense mechanisms [17]. In recent years, there has been a growing emphasis on the utilization of environmentally friendly approaches, coupled with the suggestion of integrating them with nanotechnology [18]. Micronutrients such as zinc, copper, and iron play vital roles in enzyme and biomolecule structures and functions [19]. However, conventional fertilizers face challenges in providing these nutrients to plants as they have lower bioavailability [20,21]. Compared to conventional fertilizers, nano-fertilizers (NFs) offer numerous advantages, including variable solubility, consistent and effective performance, controlled release over time, targeted activity with optimal concentration, and reduced eco-toxicity. NFs are readily absorbed by plants, preventing nutrient waste and minimizing environmental degradation. Additionally, NF formulations have demonstrated improved effects on crop growth and production, resulting in enhanced efficiency and sustainability [22]. Alongside the significant benefits of NFs and nanotechnology in agriculture, it is crucial to recognize and mitigate the potential drawbacks and risks associated with their implementation. In recent years, researchers have issued warnings concerning the hazards arising from the application of NFs, particularly regarding their tendency to accumulate, permeate, and rapidly concentrate within the tissues of humans and other organisms [23]. In addition to possessing the advantages of NFs, nano-chelated fertilizers also benefit from the advantages of chelated fertilizers, such as stability in soil, high solubility in water, high absorption capacity by plant roots, prevention of leaching in alkaline soils, and prevention of nutrient elements from being washed away in the soil [24].

Extensive research has been conducted on the impact of different fertilizers on the life cycle of *S. graminum*. Nitrogen (N) fertilizers have received the most significant attention in terms of study and analysis. One of the seminal studies on the influence of N fertilizer on aphids was conducted by Rodriguez [25], revealing a discernible positive response of aphids to elevated N levels in plants. Subsequently, Archer et al. [26] demonstrated that *S. graminum* infesting sorghum exhibited a favorable reaction to the N content in leaves, leading to a notable increase in its population. Similarly, they observed that the wheat aphid population on sorghum displayed a proportional rise in conjunction with escalating N levels. Furthermore, it has been established that an augmentation in N concentration corresponds to a deceleration in the population growth rate of *S. graminum*. Similar results were observed in relation to wheat aphids in a study conducted by Zarasvand and Allahyari [27] in Iran. They reported that with the increase in N concentration, the parameters of population increase in aphids became slower.

The aim of this study is to evaluate the effects of chelated NFs, iron, zinc, copper, and N fertilizer (urea) on *S. graminum* life table parameters. The present study is the first report on the effects of these NFs on the population parameters of *S. graminum*.

## 2. Materials and Methods

The research was conducted at the Experimental Plant Physiology Laboratory and research greenhouse located in the Faculty of Agriculture at the University of Tabriz (Tabriz, Iran).

### 2.1. Plant Material

Wheat seeds (cv. Chamran) were sown in 3 L pots filled with perlite. The pots were then transferred to a greenhouse with controlled environmental conditions (22 ± 1 °C, 60 ± 5% RH, and a 16:8 (L:D) h photoperiod). Tap water was used for irrigation every other day until germination, which occurred within two weeks. After germination, the plants were irrigated with Hoagland (Hog) nutrient solution [28] and treated with the tested NFs on alternate days [19].

### 2.2. Tested Nano-Materials

The NFs used in this study, including nano-zinc (nano-Zn), nano-copper (nano-Cu), and nano-iron (nano-Fe), were obtained from Sodour Ahrar Shargh Company (Tehran, Iran) and added to the nutrient solution. Figure 1 presents the electron microscope images and particle size analysis of the nano-fertilizers utilized in the study. Additionally, N in the form of urea (Merck Co., Darmstadt, Germany) was included in the nutrient solution. The concentrations of the NFs were determined based on the recommended concentrations provided by the manufacturing company, using the average of the high and low values for each pot (Table 1). The control plants received the Hog solution.

### 2.3. Insect Culture

The experimental period for constructing the aphid life table spanned approximately 41 days. The initial population utilized to establish a colony was adequate for infesting the initial plants and studying the offspring. The initial population of *S. graminum* was sourced from an existing colony at the Department of Plant Protection, University of Mohaghegh Ardabili, Iran. These aphids were specifically reared on cultivated wheat plants with three leaves. To ensure an adequate number of aphids for the experiment, infested plants were regularly replaced with healthy ones. The plants were replaced every two weeks due to the yellowing and wilting caused by aphid feeding, necessitating new plants to sustain the experiment. The number of plants tested corresponded to the number of aphids raised. Each plant was assigned one aphid. To prevent the aphids from escaping the target plants, adult aphids were confined to the leaves using a plastic straw measuring 0.5 × 10 cm with ventilation holes (Figure 2). Any newly hatched nymphs were promptly removed from the plants, allowing only the primary mother aphids to feed. The adult apterous aphids were randomly chosen for each fertilizer ratio and placed on the underside of leaves using a fine-haired brush inside specialized cages (Figure 2). After 12 h, the fecundity of the aphids was examined, and only one nymph of the first instar remained on each leaf. (Nymphoposition is a behavior in certain insect species like aphids where the female deposits her eggs near or on the nymphal stage of the same species). In contrast, the remaining nymphs and adult aphids were removed from the plants. For each treatment, 60 first-instar nymphs were introduced onto the leaves, with each plant assigned a distinct identification number. Every day at a specific time, the aphids were inspected, and their mortality and molting on the plants assigned to each number were recorded. This process continued until the last nymph reached adulthood. After the nymphs matured, the number of offspring produced by each adult aphid was recorded daily and then removed. The total number of offspring produced by the adult aphids until the death of the last aphid in each fertilizer ratio was counted.

### 2.4. Life Table

The fertility life table is a chronological table that shows the birth, age, and death of individuals. This table was created by following a cohort of individuals of the same age and recording their survival and time of death until the death of the last individual in the cohort. Population growth can be demonstrated using the fertility life table, which expresses the reproductive potential of female insects at different times [29,30].

The life tables start by measuring statistics such as the age of the individual (x), the number of survivors at the beginning of each age (l_x_), and age-specific fertility (m_x_). This means that it has a column for the age-specific life table.

The x (age of the individual) is a characteristic used for classifying individuals into age groups.

The l_x_ (number of survivors at the beginning of each age) is the ratio of individuals of the same age who have survived from birth to age x, obtained from the following equation:(1)$$l_{x}=\frac{N_{x}}{N_{0}}$$
where N_x_ is the number of individuals alive in each generation and N_0_ is the total number of individuals at time zero. The value of l_x_ always starts at one and decreases over time until it reaches zero.

The m_x_ (age-specific fertility or number of offspring produced per female at age x) is the average number of offspring produced per unit of time, indicating a point or continuous time interval from t to t + 1. In these calculations, usually only females are counted. If the sex ratio is 1:1, mx will be equal to the total number of offspring produced divided by two in each time interval.

The sex ratio refers to the ratio of females to the total number of offspring produced, and m_x_ is obtained by multiplying the total number of offspring produced by each female by the sex ratio. In the case of aphids, due to the phenomenon of parthenogenesis and the scarcity of males in the spring and summer seasons, the sex ratio is 100% female.

By forming the main columns of the fertility life table, its parameters can also be calculated. These parameters are derived from the analysis of available data. The main parameters calculated in a fertility life table include the net reproductive rate (R_0_), the intrinsic rate of population growth (r), the mean generation time (T), and the finite rate of population increase (λ). Determining r as the most important parameter of the fertility life table was first introduced in insects by Birch [31]. This method is based on the principles of human populations, which can be obtained by solving the Euler–Lotka equation (Equation (2)):(2)$$\sum_{x=W}^{x=0}L_{x}m_{x}e^{-rx}=1$$

The intrinsic rate of population growth is a quantitative tool or ecological index useful for comparing different species or populations, the response of different species to environmental conditions, and various factors such as temperature, humidity, food quality, plant morphology, and secondary plant compounds. The term “intrinsic” means that this is the only parameter that indicates the physiological characteristics of a living organism related to its ability to increase its population [32].

Other parameters of the life table are also obtained from the following equations [32]:

(GRR): Gross reproductive rate, the average total number of eggs produced (offspring) by a female during her lifetime.
(3)$$GRR=\sum m_{x}$$

(R_0_): Net reproductive rate, the number of offspring produced per female during one generation, taking into account the survival rate.
(4)$$R_{0}=\sum l_{x}m_{x}$$

(T): Mean generation time, the average duration of one generation. T is obtainable once r has been calculated previously and can be obtained using the following formula:(5)$$T=\frac{lnR_{0}}{r}$$

(λ): Finite rate of population increase, which is expressed in terms of days and obtained from the following formula:(6)$$\lambda =e^{r}$$
where e is the base of the natural logarithm and equals 2.71828.

The parameters r and λ can be used for the analysis and comparison of populations. Differences in the values of r and λ will result in differences in the number of individuals over time. If r < 0 or λ < 1, the population will decrease over time. If r > 0 or λ > 1, the population will increase over time. If r = 0 or λ = 1, the population will remain unchanged [33].

### 2.5. Statistical Analysis

The Chi method and Twosex-MSChart software (version 2019) were used to analyze the life tables of wheat aphids. All data related to the life tables of individuals and their life stages were analyzed using the two-sex life tables method [30]. One feature of this method is that, unlike other methods that only adjust the life table based on the female sex, it also takes into account the male sex and individuals who die before reaching maturity in the statistical analysis [29]. The comparison of the mean and standard error of life table parameters was calculated using the bootstrap method with 100,000 iterations (paired bootstrap).

## 3. Results

### Life Table Parameters of Schizaphis graminum

The average duration of immature life stages of the green wheat aphid on wheat plants treated with different fertilizers is presented in Table 2. Among the applied treatments (n = 60, *p*-value < 0.05), the longest duration of the first instar nymph stage was seen in the nano-Cu treatment, while the shortest duration was revealed in the water treatment. Regarding the duration of the second instar nymph stage, the longest duration was obtained in the nano-Cu treatment, while the shortest duration was recorded in the Hog treatment. As for the third instar nymph stage, the longest nymphal age was recorded in the nano-Cu treatment, while the shortest was observed in the Hog treatment. The longest duration of the fourth instar nymph stage was viewed in the water and nano-Cu treatments, while the shortest duration was seen in the Hog treatment, and no significant difference was observed between the Hog treatment and the nano-Zn and nitrogen treatments. The longest duration of the pre-adult stage (total nymphal period) was recorded with the nano-Cu treatment, while the shortest was associated with the Hog treatment. The longest adult insect longevity was observed in the Hog and N treatments, while the shortest longevity was seen in the water and nano-Cu treatments. The shortest total longevity of the aphids was observed in the water treatment. No significant difference was observed in the total longevity of the aphids among the other fertilizer treatments.

As shown in Table 3, the nitrogen treatment exhibited the highest fecundity, representing the production of nymphs per female individual throughout a generation, whereas the nano-Cu and water treatments demonstrated the lowest fecundity. No statistically significant difference was found between the nano-Fe and Hog treatments. The N treatment also displayed the highest number of nymphs laid per day (23.95 individuals), while the water and nano-Cu treatments showed the lowest numbers (15.48 and 16.87 individuals, respectively) (*p*-value < 0.05). No significant difference was found between the Hog and nano-Fe treatments, as well as between the nano-Fe and nano-Zn treatments (*p*-value > 0.05). The pre-reproductive period of adult individuals, known as APRP, was observed to have the highest value in the water treatment (3.42 days) and the lowest value in the N treatment (1.6 days). Similarly, for the parameter TPRP (total pre-reproductive period), the highest value was observed in the nano-Cu treatments (10.94 days), while the Hog and nano-Zn treatments demonstrated the lowest values (8.09 and 8.2 days, respectively) (*p*-value < 0.05).

The fertilizer treatments showed a significant impact on the intrinsic rate of population increase (r), with the nitrogen treatment (0.2027) demonstrating the highest and the nano-Cu treatment (0.1554) exhibiting the lowest (*p*-value < 0.05). Similarly, the highest finite rate of population increase (λ) was observed in the N treatment (1.2247), while the nano-Cu treatment (1.1682) displayed the lowest. Regarding the net reproductive rate (R_0_), the Hog treatment had the highest value, while the water and nano-Cu treatments had the lowest. No significant difference was found between the nano-Cu and nano-Zn treatments related to this parameter. The longest mean generation time (T) was observed in the nano-Cu treatment (19.39 days), while the shortest duration was recorded in the nano-Zn treatment (16.9 days). Furthermore, regarding the gross reproductive rate (GRR), the nitrogen treatment had the highest value, while the nano-Cu, nano-Zn, and water treatments had the lowest. (Table 4).

During the immature stage, the water treatment had the highest mortality rate, followed by the nano-Cu treatment. The mortality rates in the pre-adult stages were almost equal for the Hog and N treatments. The nano-Fe treatment had the lowest mortality rate in the pre-adult stages (Figure 3).

The results, as depicted in Figure 4, demonstrated the impact of different fertilizer treatments on survival and the age-specific fecundity of *S. graminum*. The highest age-specific fecundity of the aphid was observed on days 19, 18, 21, 20, 20, and 14 in the nano-Fe, nano-Zn, nano-Cu, N, Hog, and water treatments, respectively. Hog and N treatments showed the highest age-specific fecundity values, and the lowest was in thenano-Cu and nano-Fe treatments. The highest m_x_ values were 1.63, 1.47, 1.43, 1.89, 2.16, and 1.61 for the nano-Fe, nano-Zn, nano-Cu, N, Hog, and water treatments, respectively. The evaluation of survival rates revealed that the treatments for nano-Fe, nano-Zn, nano-Cu, N, Hog, and water had the greatest rates, with respective values of 0.98, 0.95, 0.98, 0.98, and 0.93. Each treatment had a distinct day when nymphopositioning started; the treatments for nano-Fe, nano-Zn, nano-Cu, N, Hog, and water showed nymphopositioning on days 8, 6, 8, 6, 5, and 7, respectively.

In the principal component analysis, two primary components accounted for 89.4% of the observed variance. Principal component 1 exhibited a pronounced positive influence on N and Hoag treatments while demonstrating a notable negative impact on nano-Cu and water treatments. Conversely, principal component 2 displayed a significant positive effect, specifically on nano-Cu treatment. Principal component 1 exerted a more substantial influence on parameters such as intrinsic rate of increase (r), net reproductive rate (R_0_), fecundity, and TPRP, accounting for approximately 96%, 97%, 97%, and −85% of the variance, respectively. On the other hand, principal component 2 predominantly influenced pre-adult longevity, explaining approximately 78% of the observed variance (Figure 5).

The heat map analysis (Figure 6) revealed several significant correlations among the life table parameters of *S. graminum* across various fertilizer treatments. Specifically, adult longevity exhibited strong associations with fecundity (81%), oviposition day (83%), intrinsic rate of increase (r) (81%), net reproductive rate (R_0_) (84%), and finite rate of increase (λ) (81%). Fecundity demonstrated notable correlations with oviposition day (99%), r (89%), R_0_ (99%), λ (89%), and gross reproductive rate (GRR) (95%). Additionally, the intrinsic rate of increase (r) displayed correlations with GRR (78%), R_0_ (89%), and λ (100%). GRR exhibited correlations with R_0_ (92%) and λ (79%). Lastly, λ was found to be correlated with R_0_ (89%).

## 4. Discussion

In natural ecosystems, herbivorous insects and predators play a substantial role in the food chain. It is worth noting that heavy metals have the potential to be taken up by plants and subsequently transferred to herbivorous and predatory insects [34]. Insects, such as aphids, can experience genetic disorders, apoptosis induction, and reduced survival as a result of contamination with heavy metals [35].

Iron acts as a catalyst for the production of hydroxyl radicals, which may cause considerable harm to the integrity of cells and tissues and exhibit cytotoxic effects. Therefore, careful management of iron transit and storage is necessary to prevent its negative effects [36,37,38]. Consequently, one effective approach to perturbing the insect’s normal physiological processes involves manipulating this essential element. Iron is taken up by the intestinal epithelial cells of insects [39]. Limited research has been undertaken regarding the influence of iron fertilizer on the life cycle of insects. The impacts of iron on various insect species, particularly those that feed on plants, remain largely unexplored. Consequently, the present study holds substantial potential for enhancing our comprehension of the iron (especially nano-Fe) effects and the alterations it instigates in life table parameters, specifically about aphids. According to the research conducted by DehghaniYakhdani et al. [40], the appropriate application of a specific element like iron can enhance the developmental period of aphids. However, the use of iron fertilizer, when not properly regulated, can potentially lead to an outbreak of the pest population. In the case of the common pistachio psyllid, *Agonoscena pistaciae*, in Burckhardt and Lauterer, on pistachio trees, it was observed that fertilizing and treating iron-deficient trees resulted in a significant increase in the population of this pest. Furthermore, this led to a shift in the life table parameters, favoring the pest. The study also discovered that a crucial factor in the interaction between the plant and the psyllid is the Fe and N ratios. Moreover, the previously cited research revealed a nonlinear relationship between population growth and micronutrient levels. The outcomes of the current investigation delineate a contrary trend. Specifically, the application of nano-Fe to wheat exhibited a noteworthy reduction in both the nymphoposition period and the number of eggs in comparison to the control. Furthermore, it exerted an adverse influence on the total pre-reproductive period (TPRP) and adult pre-reproductive period (APRP), thereby extending the temporal span of these parameters. This conveys that the utilization of nano-Fe on wheat not only averts aphid infestations but also manifests a regulatory influence conducive to the management of wheat green aphid. Interactions between different micronutrients determine how micronutrients affect these parameters. Additionally, during the investigation of the combined effects of heavy elements (cobalt, zinc, tannic acid, and gallic acid) and phenolic compounds (gallic acid, tannic acid) on the larvae of *Hyphantria cunea*, in Drury, it was noted that the pupae of larvae fed diets enriched with Fe had significantly less lipid content than the larvae in the control group [41]. In a seminal study conducted by Chatterjeet et al. [42], the entomocidal properties of chemical compounds containing Co^3+^ and Fe^3+^ were investigated, specifically focusing on their ovicidal and sterilizing effects. These investigations align with our observations pertaining to the deleterious impacts of nano-iron on the life table and life cycle of aphids.

Zinc, as an important micronutrient, serves various essential functions within the bodies of living organisms. Zinc plays a crucial role in many biological functions, including reproductive processes, DNA synthesis, behavioral responses, bone structure (in vertebrates), as well as growth and wound healing [43]. According to Rizvan et al. [44], in the context of elevated cadmium levels in plants and contaminated lands, zinc (Zn) is utilized to mitigate the toxic effects of cadmium. Consequently, it is plausible that the concentration of zinc in certain heavy metal-contaminated lands, including those contaminated by cadmium, may increase, potentially leading to direct or indirect impacts on insects. Nevertheless, elevated concentrations of zinc metal can exhibit toxicity and detrimental effects on living organisms. Extensive research has demonstrated that insects exert considerable adaptations to counteract the adverse impacts of heavy metal abundance. These adaptations induce alterations in their physiological states, including a reduction in lifespan, diminished reproductive capacity, heightened mortality rates, population decline, decreased body biomass, and modifications in generation timing [45,46,47]. The present study investigated the impact of nano-Zn on the longevity of wheat green aphids. Surprisingly, the total longevity of aphids was not affected by the nano-Zn treatment. Interestingly, the effects of nano-Zn treatment on the TPRP and the APRP were less pronounced compared to the effects of nano-Fe treatment. Moreover, the nano-Zn treatment led to a reduction in the number of nymph generation days, which is a desirable outcome from a control perspective. The value of R_0_ was lower in the nano-Zn compared to the control. This finding suggests that zinc could serve as a significant option for integrated pest control strategies. In terms of the GRR parameter, the nano-Zn treatment yielded a significantly low value, comparable to the effects observed with nano-Cu treatment and elemental deficiency treatment (water treatment). Consequently, these findings suggest that combining nano-Zn treatment with other methods could be a promising option for integrated pest management (IPM) strategies. It is well-studied that the main effect of zinc accumulation in the insect’s body is toxicity, which leads to reproductive disorders [48]. In a study conducted by Xi et al. [34], which focused on *Aphis medicaginis* and its predator, the ladybird *Harmonia axyridis*, it was seen that exposure to zinc had a detrimental effect on aphid reproduction, primarily due to alterations in the expression of the vitellogenin coding gene. Consequently, the predation of zinc-exposed aphids by ladybirds decreased, resulting in a decline in the number of eggs laid by the ladybirds. These findings highlight the role of zinc metal in influencing the expression of the vitellogenin gene, thereby diminishing the reproductive capacity of both aphids and ladybirds. Our current research further supported these findings, as we observed a significant reduction in the reproductive rate following treatment with nano-Zn, aligning with the aforementioned studies. In the study [49] on *Spodoptera litura*, it was found that the high amount of zinc had a negative effect on the quality of the eggs, which is due to the effect that excess zinc has on the energy reserves of the insect [48], and this can be the most important factor in disrupting egg laying and larval development. Studies by Kafel et al. [50] on the toxic effects of zinc and cadmium on *Spodoptera exigua* clearly showed that exposure of the insect to these elements significantly reduces its survival. In our previous investigation [19] on the impact of nano-Zn treatment on wheat plants and wheat green aphid, we discovered that the application of nano-Zn led to an increase in the phenol content within aphid-infested wheat plants. This rise in phenol levels can be interpreted as an inhibitory factor against *S. graminum*, the aphid in question. Furthermore, the study revealed that the presence of nano-Zn resulted in relatively elevated levels of catalase (CAT) and superoxide dismutase (SOD) in the aphid. These findings indicate the toxic effects of nano-Zn on this particular insect species. Consequently, it can be inferred that the relatively limited impact of nano-Zn on the aphid’s life table in this study may be attributed to the activation of the aphid’s CAT and SOD enzymes, which have partially detoxified the toxicity associated with nano-Zn exposure.

Copper (Cu^2+^) is one of the seven essential micronutrients for insects’ growth and reproduction and is considered an important component of many enzymes and proteins in cells [51]. Copper concentrations within a certain range have a catalytic influence on the growth, reproduction, and development of insects; they can also improve weight and body size [52,53]. In low concentrations, Cu can reduce the development time of insects [54]. However, higher concentrations of copper exhibit irreversible toxicity toward insects [55]. The rise in Cu levels in the environment can be attributed to human activities [56]. For instance, the application of Bordeaux’s solution for managing fruit tree rot disease leads to a significant increase in Cu levels in the soil [57,58]. Consequently, insects feeding on Bordeaux’s solution-treated leaves led to enhanced reproduction and even pest outbreaks [58]. On the other hand, mortality rises and population development is negatively impacted when insects feed on sources with greater concentrations of Cu, as has been reported in *Boettcherisca peregrina* [52] and *Apis mellifera* L. [59]. Furthermore, it has been established that an elevated level of copper ions results in significant harm to the insect’s midgut [60]. The damaging impacts of the elevated copper concentration on the midgut’s microbial flora lead to a decline in the insect’s ability to survive and disrupt its development [61]. In investigations concerning the impact of elevated copper concentrations, *Proisotoma minuta* demonstrated elevated mortality, slow development, and diminished reproduction in jars holding the soil with the maximum concentration of Cu [62]. The results reported above align with our findings on Cu, which demonstrated that water and nano-Cu treatments decreased the aphids’ reproduction rate. In a study conducted by Yang et al. [63] on *S. litura*, the results showed that not only the survival rate of immature stages but also the intrinsic rate of population increase (r) and the finite rate of population increase (λ) significantly increased under lower Cu concentrations (2, 4, and 8 mg/kg). Furthermore, the population growth of *S. litura* was impressively faster, indicating the insect’s ability to adapt to low concentrations and possibly outbreaks. However, at high Cu concentrations (32 mg/kg), there was also a substantial reduction in the survival rate of the immature stages, as well as in the rate of population increase, pupa weight, length, adult insect body weight, and egg laying. The adult stage’s survival rate was extremely poor when the dose reached 64 mg/kg, demonstrating that *S. litura*’s adaptability had decreased. The Cu concentration utilized in this study is potentially detrimental since it resulted in the lowest intrinsic rate of population increase (r) and finite rate of population increase (λ), indicating a negative influence on these parameters. Numerous studies have indicated that high Cu concentrations prolong the larval development period of *S. liuta*, *Aedes aegypti* L., and *B. peregrina*, likely due to a drop in emergence- and appearance-related protein levels [52,60,64]. In the present study, the length of the pre-pubertal period was significantly higher in the nano-Cu treatment than in all other treatments. This suggests that nymphs of different ages of the aphid are more sensitive to Cu metal at this concentration and are strongly affected by it. According to Yang et al. [63], high Cu concentrations in *S. litura* caused a decrease in the reproduction rate and an increase in TPOP and APOP. The concentration of metal ions was identified as the source of this decline [65]. Furthermore, Shu et al. [49] showed that suppression or reduction of vitellogenin (Vg) synthesis results in lower fertility and reproduction. Zhou et al. [66] found a similar relationship between these variables and a lower fertility rate. Like Cu, zinc metal has been related to decreased egg yolk protein reserves and changed vitellogenin gene expression, which has decreased fertility and reproduction [49]. These outcomes were nearly in line with our findings, which showed that in the case of aphids, the nano-Cu treatment led to the most TPRP and the least amount of reproduction in terms of element abundance. The results of our previous investigation [19] regarding the impact of nano-Cu treatment on wheat and *S. graminum* demonstrated that this treatment did not exhibit significant toxicity toward wheat, only moderately affecting its enzymatic and non-enzymatic antioxidant systems. Conversely, the aphid displayed notably elevated levels of peroxidase (POX), CAT, and SOD enzymes, suggesting the pronounced toxicity of nano-Cu on the aphid. Analysis of the life table and examination of the aphid’s antioxidant systems consistently revealed the substantial toxicity exerted by nano-Cu, as evidenced by both the antioxidant system and aphid life table parameters.

Nitrogen is an essential factor that increases aphid populations. Aphids respond positively to an increase in nitrogen. For example, a study by Alasvand et al. [67] investigated the effect of different nitrogen fertilizer levels on wheat aphids and found that nitrogen fertilizer increased the aphid’s intrinsic rate of population increase (r) from 0.246 to 0.267 per day as nitrogen concentration increased from 0 to 150%. This study also showed that aphid reproduction increased with nitrogen, a finding that was also supported by our study, in which the nitrogen treatment exhibited the highest aphid reproduction among all treatments. The intrinsic rate of population increase in our study was lower than the value reported by Alasvand et al. [67] because we added nitrogen to Hoagland’s nutrient solution, which already contains all the elements and nutrients that plants need. The interaction of nutrients in Hoagland’s solution with nitrogen should also be considered. Nevertheless, in our study, aphids grown on wheat treated with nitrogen fertilizer showed the highest intrinsic rate of population increase (0.2027/day) among all treatments. Numerous studies have been conducted on the relationship between nitrogen fertilizer application and increased reproduction and population size of various aphid species, both in the field and in the laboratory. The results have consistently shown that aphid populations increase with increasing nitrogen levels [26,68,69,70]. Dorshner et al. [71] showed that wheat aphids can alter plant metabolism and induce senescence in wheat leaves. Some biotypes of wheat aphids cannot alter plant metabolism and therefore have lower reproductive capacity than other biotypes. Our previous investigation [19] concerning the influence of nitrogen on wheat plants and *S. graminum* aphids revealed notable variations in the levels of antioxidant enzymes (POX, CAT, and SOD). Specifically, the wheat plants exhibited elevated levels of these enzymes, while the aphids displayed comparatively lower enzyme levels. This suggests that nitrogen does not confer significant resistance to the plant, and the stress experienced by the aphids in these plants induces heightened enzyme activity. Conversely, the diminished levels of antioxidant enzymes in the aphids indicate their affinity for higher nitrogen concentrations, implying that aphids thrive in nitrogen-rich environments. Consequently, the life table parameters of the aphids demonstrated the highest values under nitrogen treatment in comparison to other treatments.

Regarding the negative effect of water treatment on reproduction and other parameters of the aphid’s life table, we can mention the food quality of plants treated with water (only water as a nutrient deficiency), which is at a lower level in terms of quality level and nutrients and micronutrients, especially in plants and aphids. This deficiency showed its effect more, because in the water treatment, except for the soluble elements, due to the percentage of water hardness, no other element was available to the plant. Altering the life table parameters in a manner that is unfavorable to the insect, such as experiencing lower longevity in adult aphids, shorter total longevity, the lowest level of fecundity among the treatments, the highest APRP, a high TPRP, a low intrinsic rate of population increase (r), the lowest R_0_, and the highest pre-adult mortality rate, suggests that this treatment is not suitable for the aphid. Therefore, it is possible to take into account the higher probability of a lack of nutrients and the low quality of food in water treatment. In this study, the water treatment (representing element deficiency) was found to be detrimental to both the plant and the aphid. This treatment was included for comparison purposes, highlighting that element deficiencies can have similar consequences to specific fertilizer treatments for aphids and other pests. Furthermore, element deficiencies can negatively impact plant health.

## 5. Conclusions

The findings of this research, which focused on the life table of the wheat aphid (*Schizaphis graminum*), revealed that the nano-Cu treatment had a significant negative impact on the aphid’s life table parameters. This result suggests that incorporating nano-Cu-enhanced plant nutrition could be an effective strategy for integrated pest management of the aphid, as it improves plant nutrition and alters population parameters. The nano-Fe and nano-Zn treatments also exhibited negative effects on the aphid, although to a lesser extent compared to the nano-Cu treatment. However, these fertilizers demonstrated specific impacts on the life table parameters. By employing appropriate ratios of these fertilizers or a combination of all three, taking into account plant physiology and the adverse effects on the aphid’s life table, it may be possible to achieve effective integrated pest management. Considering the existing literature on the reduction in environmental pollution, NFs hold promise for sustainable agriculture. Nano-fertilizers have become a valuable asset in controlling aphids, offering improved plant growth and health through their specialized properties. These fertilizers, consisting of nano-sized particles, enhance nutrient absorption and targeted delivery to plants, ultimately bolstering plant defenses against aphid infestations. By providing essential nutrients more effectively, nano-fertilizers help plants resist aphids while preventing excessive vegetative growth that could attract them. In summary, nano-fertilizers play a crucial role in sustainable aphid management by promoting plant health and reducing vulnerability to aphid attacks.

In conclusion, there is no one-size-fits-all approach to the use of NFs in agriculture and pest control. It is crucial to conduct comprehensive studies that encompass various fields of agriculture and environmental sciences, such as agronomy, horticulture, soil science, water and food pollution, and the impact of these substances on human health and the economic viability of this technology. These studies will provide valuable scientific recommendations for the effective and responsible use of different NFs in plant nutrition, pest and disease management, and their overall implications.

## Figures and Tables

**Figure 1 insects-15-00209-f001:**
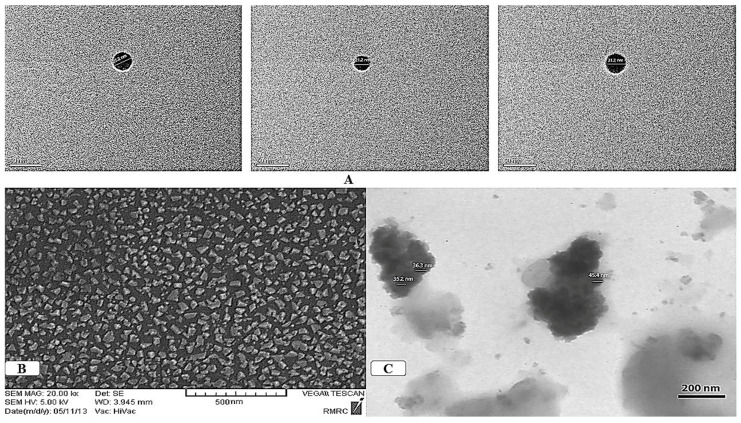
The TEM images of nano-chelated fertilizers: (**A**) Fe nano-chelated fertilizer; (**B**) Cu nano-chelated fertilizer; (**C**) Zn nano-chelated fertilizer (Note: all of these pictures were prepared by the Sodour Ahrar Shargh Company).

**Figure 2 insects-15-00209-f002:**
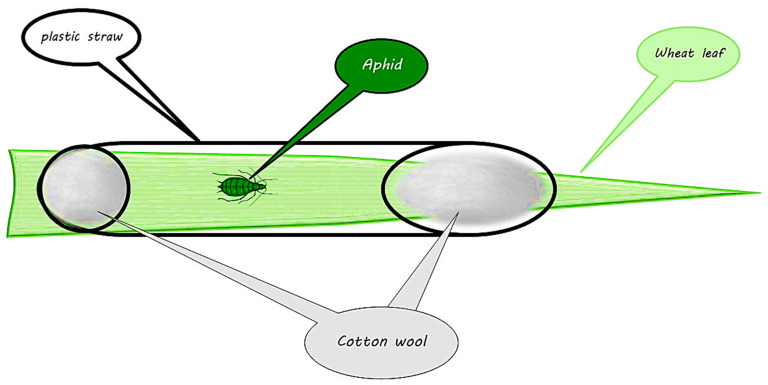
Schematic depiction of the method used to enclose aphids on wheat plant leaves for conducting life table experiments.

**Figure 3 insects-15-00209-f003:**
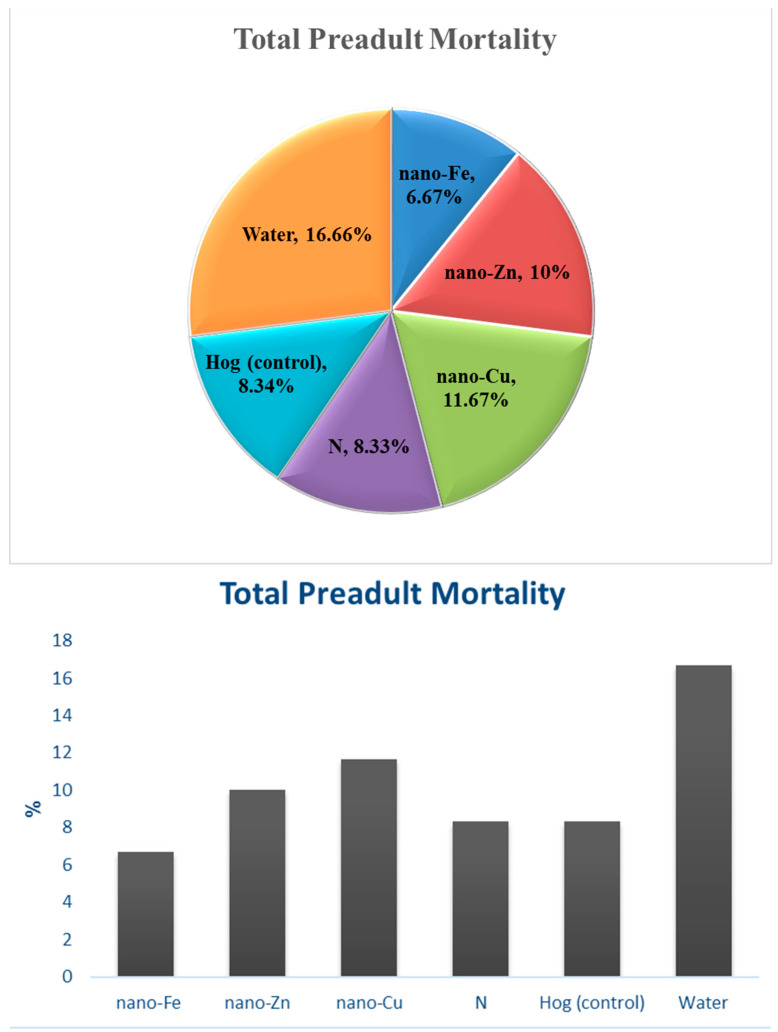
Total pre-adult mortality in *Schizaphis graminum* reared in wheat plants treated by nitrogen, nano-fertilizers (nano-Cu, -Zn, and -Fe), and water.

**Figure 4 insects-15-00209-f004:**
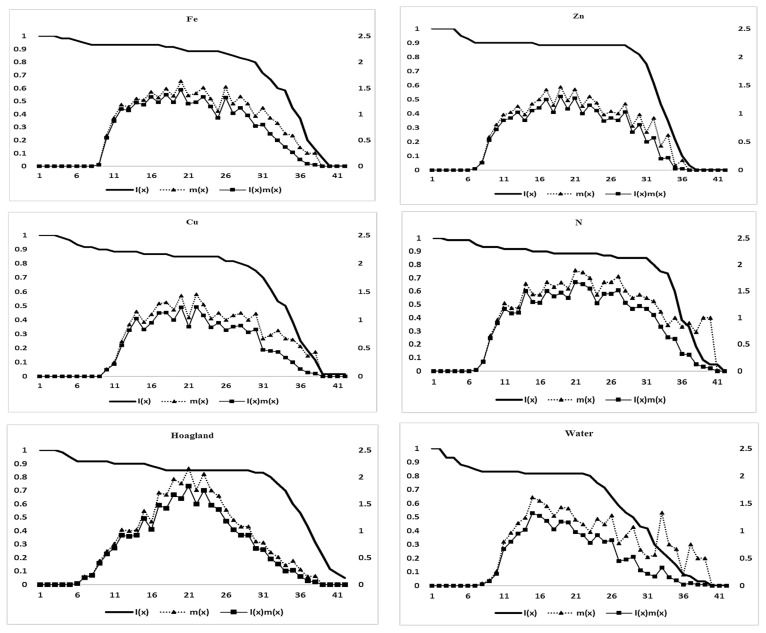
Age-stage survival rate (l_x_), age-specific fecundity (m_x_), and maternity (l_x_m_x_) of *Schizaphis graminum* reared on the treated wheat plants with nitrogen, different nano-fertilizers, and water.

**Figure 5 insects-15-00209-f005:**
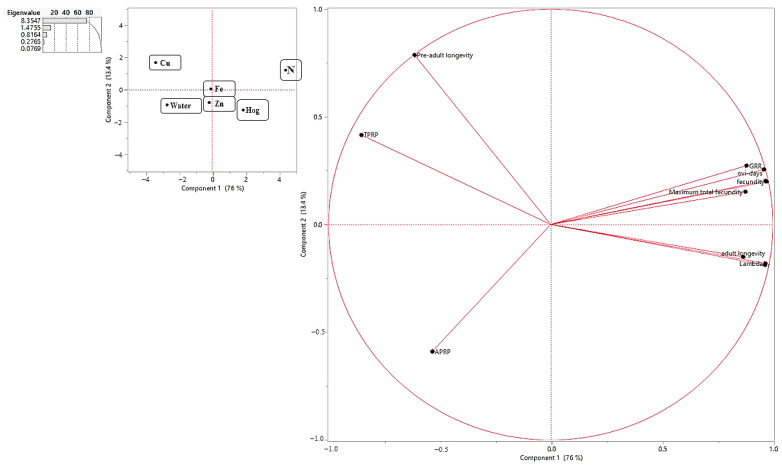
PCA based on correlation among life table parameters of reared *Schizaphis graminum* on wheat plants treated with nitrogen, nano-fertilizers (nano-Cu, -Zn, and -Fe), and water.

**Figure 6 insects-15-00209-f006:**
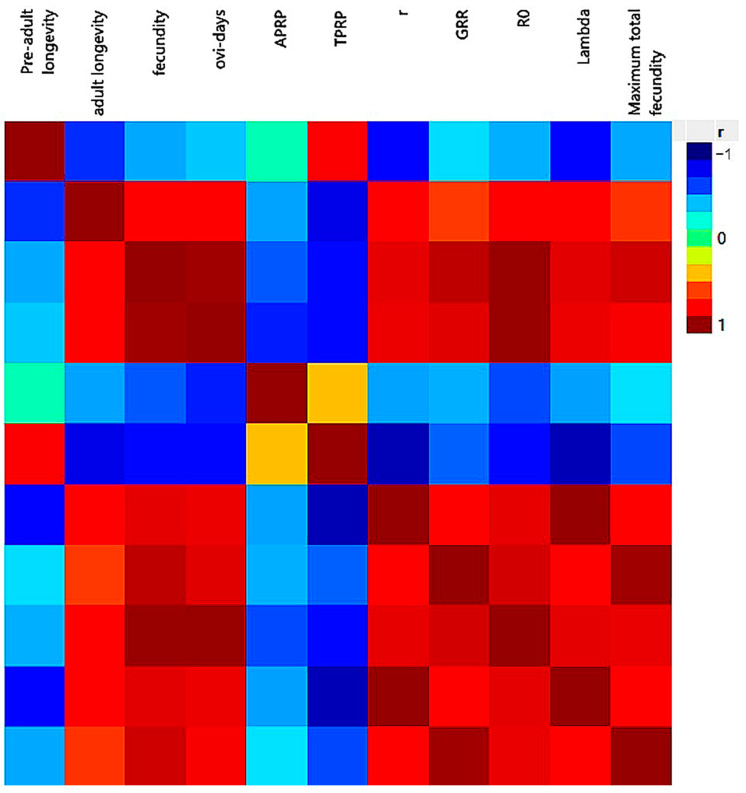
Heat map on the correlation between life table parameters of *Schizaphis graminum* reared on wheat plants treated with nitrogen, nano-fertilizers (nano-Cu, -Zn, and -Fe), and water.

**Table 1 insects-15-00209-t001:** The recommended fertilizer application instructions for pot cultivation include the manufacturing company’s instructions and the corresponding amount of fertilizer to be used per pot.

Fertilizers	N (Urea)	Zn (Chelated Zinc 12%)	Cu (Chelated Copper 8%)	Fe (Chelated Iron 9%)
Manufacturing company instruction (kg/ha)	100	4–6	1–2	3–10
The amount used per pot (mg)	83.3	6.25	1.87	8.12

**Table 2 insects-15-00209-t002:** Duration of developmental time (day) (mean ± SE) of different stages of *Schizaphis graminum* reared on wheat leaves treated with nitrogen and different nano-fertilizers.

Stage (Day)	Nano-Fe	Nano-Zn	Nano-Cu	Nitrogen (N)	Hoagland (Control)	Water
Nymph 1	1.67 ± 0.075 ^b^	1.4 ± 0.065 ^c^	2.11 ± 0.074 ^a^	1.82 ± 0.056 ^b^	1.35 ± 0.062 ^c,d^	1.19 ± 0.053 ^d^
Nymph 2	1.81 ± 0.073 ^b^	1.62 ± 0.069 ^b,c^	2.15 ± 0.105 ^a^	1.57 ± 0.07 ^c^	1.19 ± 0.053 ^d^	1.45 ± 0.068 ^c^
Nymph 3	1.61 ± 0.07 ^b,c^	1.54 ± 0.073 ^c^	2.13 ± 0.103 ^a^	1.51 ± 0.067 ^c^	1.25 ± 0.058 ^d^	1.74 ± 0.061 ^b^
Nymph 4	1.84 ± 0.061 ^b^	1.69 ± 0.069 ^b,c^	2.4 ± 0.087 ^a^	1.67 ± 0.064 ^b,c^	1.58 ± 0.067 ^c^	2.22 ± 0.059 ^a^
Adult	26.25 ± 0.633 ^b^	26.19 ± 0.455 ^b^	24.17 ± 0.692 ^c^	28.02 ± 0.644 ^a^	29.36 ± 0.871 ^a^	23 ± 0.672 ^c^
Preadult	6.93 ± 0.091 ^b^	6.24 ± 0.079 ^d^	8.77 ± 0.139 ^a^	6.53 ± 0.077 ^c^	5.42 ± 0.126 ^e^	6.62 ± 0.124 ^c^
Total longevity	31.23 ± 1.128 ^a^	29.58 ± 1.182 ^a^	29.57 ± 1.349 ^a^	32.18 ± 1.187 ^a^	32.23 ± 1.362 ^a^	25.28 ± 1.399 ^b^

Note: Different letter demonstrates significant differences between treatments.

**Table 3 insects-15-00209-t003:** Adult pre-reproduction period (APRP) and total pre-reproduction period (TPRP), fecundity, and nymphoposition period (mean ± standard error) of *Schizaphis graminum* reared on wheat leaves treated with nitrogen and different nano-fertilizers.

Parameters	Nano-Fe	Nano-Zn	Nano-Cu	N	Hog (Control)	Water
Fecundity (nymph)	28.36 ± 0.88 ^b^	25.61 ± 0.55 ^c^	23.09 ± 0.81 ^d^	37.35 ± 1.07 ^d^	30.65 ± 1.03 ^b^	22.72 ± 0.87 ^d^
APRP (day)	2.62 ± 0.087 ^b^	1.96 ± 0.099 ^c^	2.17 ± 0.089 ^c^	1.6 ± 0.08 ^d^	2.67 ± 0.12 ^b^	3.42 ± 0.13 ^a^
TPRP (day)	9.55 ± 0.11 ^c^	8.2 ± 0.13 ^d^	10.94 ± 0.15 ^a^	8.13 ± 0.11 ^d^	8.09 ± 0.2 ^d^	10.04 ± 0.17 ^b^
Nymphoposition days (day)	19.62 ± 0.56 ^b,c^	18.35 ± 0.42 ^c^	16.87 ± 0.56 ^d^	23.95 ± 0.62 ^a^	20.35 ± 0.7 ^b^	15.48 ± 0.54 ^d^
Maximum total fecundity (nymph)	36	32	31	51	41	36

Note: Different letter demonstrates significant differences between treatments.

**Table 4 insects-15-00209-t004:** Population parameters (mean ± standard error) of *Schizaphis graminum* reared on wheat leaves treated with nitrogen and different nano-fertilizers.

Parameter	Nano-Fe	Nano-Zn	Nano-Cu	N	Hog (Control)	Water
r (1/day)	0.1821 ± 0.0027 ^b^	0.1856 ± 0.0036 ^b^	0.1554 ± 0.0035 ^d^	0.2027 ± 0.0034 ^a^	0.1896 ± 0.0038 ^b^	0.1683 ± 0.0045 ^c^
λ (1/day)	1.1997 ± 0.0032 ^b^	1.2039 ± 0.0043 ^b^	1.1682 ± 0.0041 ^d^	1.2247 ± 0.0041 ^a^	1.2088 ± 0.0046 ^b^	1.1833 ± 0.0053 ^c^
R_0_ (Female/Gen)	26.47 ± 1.22 ^b^	23.05 ± 1.11 ^c^	20.40 ± 1.19 ^cd^	34.24 ± 1.65 ^a^	28.1 ± 1.44 ^b^	18.93 ± 1.31 ^d^
T (day)	17.99 ± 0.19 ^b^	16.90 ± 0.17 ^d^	19.39 ± 0.20 ^a^	17.43 ± 0.18 ^c^	17.59 ± 0.27 ^b,c^	17.49 ± 0.29 ^b,c,d^
GRR (female/s)	31.46 ± 0.47 ^c^	27.23 ± 0.37 ^d^	27.05 ± 0.84 ^d^	42.87 ± 1.09 ^a^	33.16 ± 0.54 ^b^	29.21 ± 0.99 ^d^

Note: Different letter demonstrates significant differences between treatments.

## Data Availability

The data presented in this study are available on request from the corresponding author.

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
