# Peer review of "Examining Innovative Technologies: Nano-Chelated Fertilizers for Management of Wheat Aphid (Schizaphis graminum Rondani)"

_insects, 2024, doi:10.3390/insects15030209_

Round 1

Reviewer 1 Report

Comments and Suggestions for Authors

You have done an excellent study on the effecs of nanofertilzers on the wheat aphid and enhanced our knowledge on the effects of these chemicals on insects generally. 

More specifically, i have various comments and suggestions:

I know that the term greenbug is used in some scientific literature, but it is such a  laymen term. They  use bugs as a generic term for all insects.Thus sick to green wheat aphid or the scientific name Check the title and lines 24,31,35,42,49,57,95,98,222,83,499 and 529.

The title would read better as ....Fertilizers  for Management of Wheat Grain Aphid...  

Line25. Write:..pest of wheat plants that can transmit

26-28. A treatment cannot "show" longevity etc. Perhaps "resulted in"

28. "led to" is OK

47. Meaning of "well defined" not clear to me

55. date back to previous years? Perhaps ''several years"

93. favourable (favorable only used by the USA and Google)

148. Perhaps the term nymphoposition should be explained here

Fig. 2 .Cannot see a straw and there is no mention of it in the text

236. Hog treatment? Not mentioned in the text

285-312. Hoag? Not used before

297-304. Now you even use Haogland! Be consistent.

353. Favouring

400-403 etc use ladybird beetle or coccinellid

414. give reference after "previous study" here (as well)

418. If "''our study" refers to previous study, write: "the study"

414-423.  I assume this is all about the previous study. Consider adding in 422 ....aphid's life table in this study"

439. Why the full reference here? Also give the reference number.

440. Short sentence non sensical

511-514. Review this  sentence  and the next one

516-521.Review long sentence---not clear

524. Write: this treatment was included  for comparison purposes

531. "result" rather than ''point"

References": I do  not like the use of capital letters for species names.The scientific names should preferably be in italics. Not sure about editorial policy.

Author Response

Comments and Suggestions for Authors

You have done an excellent study on the effecs of nanofertilzers on the wheat aphid and enhanced our knowledge on the effects of these chemicals on insects generally. 

More specifically, i have various comments and suggestions:

I know that the term greenbug is used in some scientific literature, but it is such a laymen term. They use bugs as a generic term for all insects.Thus sick to green wheat aphid or the scientific name Check the title and lines 24,31,35,42,49,57,95,98,222,83,499 and 529.

Thank you for your kind attention and excellent guidance; this term has been changed to "wheat aphid".

The title would read better as ....Fertilizers  for Management of Wheat Grain Aphid...  

Your opinion has been taken into consideration.

Line25. Write:..pest of wheat plants that can transmit

It was changed (line 25)

26-28. A treatment cannot "show" longevity etc. Perhaps "resulted in"

It was changed (line 27)

  1. "led to" is OK

Thanks

  1. Meaning of "well defined" not clear to me

It was changed to: The reproductive potential of this aphid is clearly defined and substantial; nevertheless, when subjected to biotic or abiotic pressures, this potential may display notable variations. (line 48)

  1. date back to previous years? Perhaps ''several years"

In order to avoid repetition, "several years" was used instead of "past years." (line 58)

  1. favourable (favorable only used by the USA and Google)

It was changed (line 96)

  1. Perhaps the term nymphoposition should be explained here

Thank you for your attention to this matter. I am unsure if I am allowed to add a descriptive section about terminologies in the Materials and Methods section, but based on your suggestion, this definition has been included. (line 147)

Fig. 2 .Cannot see a straw and there is no mention of it in the text

The previous Fig has been replaced with a newer one that offers more details.

  1. Hog treatment? Not mentioned in the text

In line 117, the abbreviation "Hog" was substituted for the term "Hoagland nutrient solution."

285-312. Hoag? Not used before

It was edited (line 296)

297-304. Now you even use Haogland! Be consistent.

All of them were edited (308-315)

  1. Favouring

It was edited (365)

400-403 etc use ladybird beetle or coccinellid

All of the ladybugs changed to ladybirds (413-419)

  1. give reference after "previous study" here (as well)

It was done

  1. If "''our study" refers to previous study, write: "the study"

It was changed (427)

414-423.  I assume this is all about the previous study. Consider adding in 422 ....aphid's life table in this study"

Your comment was respectfully considered, and the changes were implemented. (431-435)

  1. Why the full reference here? Also give the reference number.

It was corrected (452)

  1. Short sentence non sensical

With thanks for your careful consideration, this mistake has been rectified. (454)

511-514. Review this sentence and the next one

It was rephrased (532)

516-521.Review long sentence---not clear

Edited and enhanced

  1. Write: this treatment was included for comparison purposes

It was done (543)

  1. "result" rather than ''point"

It was changed (551)

References": I do not like the use of capital letters for species names.The scientific names should preferably be in italics. Not sure about editorial policy.

All scientific names were reviewed and corrected to be italicized.

Reviewer 2 Report

Comments and Suggestions for Authors

Journal: Insects

Manuscript ID: insects-2894764

Title: Examining Innovative Technologies: Nano-Chelated Fertilizers for Wheat

Green Bug (Schizaphis graminum Rondani) Management

Authors: Masoud Chamani *, Bahram Naseri *, Hooshang Rafiee-Dastjerdi, Javid Emaratpardaz, Reza Farshbaf Pourabad, Ali Chenari Bouket, Tomasz Oszako, Lassaad Belbahri

Abstract: The study focused on the effect of wheat leaves treated with nano-chelated fertilizers and nitrogen (N) fertilizer on the wheat green bug (Schizaphis graminum Rondani), a harmful pest in wheat plants that transmits dangerous viruses. The nano-Cu treatment showed the longest pre-adult longevity. The authors demonstrated that the nano-Cu treatment showed the lowest adult longevity, fecundity, nympho-position day’s number, intrinsic rate of population growth (r), finite rate of population increase (λ), and net reproductive rate (R0). The N treatment led to the highest levels of fecundity, nympho-position days, r, λ, and R0. Nano-Fe and nano-Zn demonstrated fewer negative effects on S. graminum life table parameters than nano-Cu. The author’s results indicate that N treatment yielded numerous advantageous effects on the wheat green bug while simultaneously impeding the efficacy of the aphid control program. Conversely, nano-Cu treatment exhibited a detrimental influence on various parameters of the aphid's life table, resulting in a reduction of the pest's fitness. Consequently, the integration of nano-Cu should be seriously considered as a viable option in the IPM of the wheat green bug.

Positive aspects:

·         Propose subject of study is very interesting and has practical implications in agriculture field.

·         The information presented in the manuscript is original and modern (focusing on one of the most important agriculture area- pest management).

·         The manuscript is sustained by a suitable and diverse literature, strongly connected with the proposed research area.

·         The manuscript is well structured.

·         The objective of the study is clear defined.

·         The methodology is detailed described, scientific argued, adequately for the proposed research area.

·         Results and discussion are detailed presented (the authors making a comparison with data from other studies, on other species).

·         The figures and tables are proper, reflecting the obtained results.

·         The proposed research subject is proper for Insect Journal.

 Comments:

·         Abstract: line 28: please insert the T and GRR parameters.

·         Introduction: Line 102. The authors declared: “The present study is probably the first report….”. The author must be certain. They must to analyze all literature on this research subject and to declare for sure which is the studies’ novelty.

·         Material and methods. Line 105: which was the period of study?

·         Figure 1: Line 128. The text under the figure 1 B is incomprehensible. Please change it.

·         Insect culture. Line 133. For how long you cultivated the plants? Which was the culture period? How many plants were used? How many aphids do you obtained?

            How many individuals of aphids did you used at the begging of the experiment?

·         Life table. Lines 218, 219: please insert some reference.

·         On figure 4, line 292: the ordinate parameters are too small. Please, change the font.

·         Conclusions. I strongly recommend highlighting the practical importance of this study.

·         References: Please, check if the all references were found in the manuscript and vice versa. Please, follow the instruction for the authors for references.

ALL these comments were inserting in the manuscript!

Author Response

Positive aspects:

  • Propose subject of study is very interesting and has practical implications in agriculture field.
  • The information presented in the manuscript is original and modern (focusing on one of the most important agriculture area- pest management).
  • The manuscript is sustained by a suitable and diverse literature, strongly connected with the proposed research area.
  • The manuscript is well structured.
  • The objective of the study is clear defined.
  • The methodology is detailed described, scientific argued, adequately for the proposed research area.
  • Results and discussion are detailed presented (the authors making a comparison with data from other studies, on other species).
  • The figures and tables are proper, reflecting the obtained results.
  • The proposed research subject is proper for Insect Journal.

 Thank you for your thorough review and valuable suggestions for improving the article. I appreciate your kind feedback and praise of the article.

 Comments:

  • Abstract: line 28: please insert the T and GRR parameters.

Thanks for reminding me of this point, it was inserted (line 29)

  • Introduction: Line 102. The authors declared: “The present study is probably the first report….”. The author must be certain. They must to analyze all literature on this research subject and to declare for sure which is the studies’ novelty.

We declared it: The present study is probably the first report on the effects of these NFs on the popula-tion parameters of S. graminum. (line 106)

The reason "probably" has been used is that similar articles to this one may be published in other parts of the world from the time of submission and peer review to acceptance and publication. However, what we know is that such a study has not been conducted at present.

  • Material and methods. Line 105: which was the period of study?

Thank you for your inquiry, but it is a bit unclear to me. If you are referring to the duration of the study period, it was part of Masoud Chamani's doctoral thesis research. The study involved various stages, such as measuring plants, aphids, and ladybugs antioxidants, as well as constructing life tables for aphids and ladybugs. These assessments were completed nearly two years ago.

  • Figure 1: Line 128. The text under the figure 1 B is incomprehensible. Please change it.

It was changed. (line 132)

  • Insect culture. Line 133. For how long you cultivated the plants? Which was the culture period? How many plants were used? How many aphids do you obtained?

            How many individuals of aphids did you used at the begging of the experiment?

The experimental period for constructing the aphid life table spanned approximately 41 days. The initial population utilized to establish a colony was adequate for infesting the initial plants and studying the offspring. Sixty (60) aphids were allocated to each treatment, as specified in the manuscript. The plants were replaced every two weeks due to the yellowing and wilting caused by aphid feeding, necessitating new plants to sustain the experiment. The number of plants tested corresponded to the number of aphids raised. Each plant was assigned one aphid.

  • Life table. Lines 218, 219: please insert some reference.

Thanks for your attention, it was added (line 229)

  • On figure 4, line 292: the ordinate parameters are too small. Please, change the font.

Thank you for your consideration, it was done (line 302)

  • Conclusions. I strongly recommend highlighting the practical importance of this study.

It was added: (line 561)

Nano fertilizers have become a valuable asset in controlling aphids, offering improved plant growth and health through their specialized properties. These fertilizers, consisting of nano-sized particles, enhance nutrient absorption and targeted delivery to plants, ultimately bolstering plant defenses against aphid infestations. By providing essential nutrients more effectively, nano fertilizers help plants resist aphids while preventing excessive vegetative growth that could attract them. In summary, nano fertilizers play a crucial role in sustainable aphid management by promoting plant health and reducing vulnerability to aphid attacks.

  • References: Please, check if the all references were found in the manuscript and vice versa. Please, follow the instruction for the authors for references.

It was checked

ALL these comments were inserting in the manuscript!